# Analysis of Speech Features in Alzheimer’s Disease with Machine Learning: A Case-Control Study

**DOI:** 10.3390/healthcare12212194

**Published:** 2024-11-04

**Authors:** Shinichi Noto, Yuichi Sekiyama, Ryo Nagata, Gai Yamamoto, Toshiaki Tamura

**Affiliations:** 1Department of Rehabilitation, Niigata University of Health and Welfare, Niigata 950-3198, Japan; toshiaki-tamura@nuhw.ac.jp; 2Mizuho Research & Technologies, Ltd., Tokyo 101-8443, Japan; yuichi.sekiyama@mizuho-rt.co.jp (Y.S.); ryo.nagata@mizuho-rt.co.jp (R.N.);

**Keywords:** Alzheimer’s disease, voice, spectrum, machine learning, artificial intelligence (AI)

## Abstract

Background: Changes in the speech and language of patients with Alzheimer’s disease (AD) have been reported. Using machine learning to characterize these irregularities may contribute to the early, non-invasive diagnosis of AD. Methods: We conducted cognitive function assessments, including the Mini-Mental State Examination, with 83 patients with AD and 75 healthy elderly participants, and recorded pre- and post-assessment conversations to evaluate participants’ speech. We analyzed the characteristics of the spectrum, intensity, fundamental frequency, and minute temporal variation (∆) of the intensity and fundamental frequency of the speech and compared them between patients with AD and healthy participants. Additionally, we evaluated the performance of the speech features that differed between the two groups as single explanatory variables. Results: We found significant differences in almost all elements of the speech spectrum between the two groups. Regarding the intensity, we found significant differences in all the factors except for the standard deviation between the two groups. In the performance evaluation, the areas under the curve revealed by logistic regression analysis were higher for the center of gravity (0.908 ± 0.036), mean skewness (0.904 ± 0.023), kurtosis (0.932 ± 0.023), and standard deviation (0.977 ± 0.012) of the spectra. Conclusions: This study used machine learning to reveal speech features of patients diagnosed with AD in comparison with healthy elderly people. Significant differences were found between the two groups in all components of the spectrum, paving the way for early non-invasive diagnosis of AD in the future.

## 1. Introduction

Alzheimer’s disease (AD) is a common age-related degenerative disease and represents a major public health concern in countries with aging populations. Reviews of AD epidemiology have reported that 2.0 to 16.8 new cases of AD occur per 1000 person-years, on the basis of studies conducted in the United States, Europe, Japan, and China [1]. Furthermore, the number of dementia-related deaths, including those from AD, more than doubled between 1990 and 2016 [2]. These reports emphasize the need for technological development for the early detection of AD.

AD is characterized by the accumulation of extracellular amyloid-β (Aβ) plaques and intracellular phosphorylated tau [3]. Diagnostic methods have expanded rapidly over the past few decades to include neuroimaging biomarkers derived from positron emission tomography (PET) of amyloid and tau, Aβ and tau analysis of cerebrospinal fluid (CSF), and more recently, plasma-based biomarkers of p-tau, total tau (t-tau), and Aβ [4].

Mild cognitive impairment (MCI) is an established precursor stage of AD that is often associated with early signs of neurodegeneration. Fluid biomarkers of MCI include elevated neurofilament light chain (NfL) and glial fibrillary acidic protein (GFAP) in CSF, as well as elevated t-tau, NfL [5], and GFAP in blood [6]. These plasma and fluid biomarkers are attractive approaches for large-scale screening efforts, given the high cost and limited availability of PET. However, the collection of plasma and body fluid samples for testing is time-consuming and costly, and researchers continue to look for easier, non-invasive screening methods.

Speech analysis is a typical example of a non-invasive testing method for evaluating speech and language function. Language function in AD is impaired in the prodromal phase, with subtle changes in syntactic complexity, and semantic and lexical content [7]. Toth et al. [8] reported that patients with MCI were distinguishable from healthy older adults in terms of speech tempo in a delayed recall task and the number of pauses in a question–answer task. Pistono et al. [9] also focused on these pauses in speech and found that they were associated with episodic memory impairment. Additionally, lexical-semantic processing was reported to be disrupted in patients with AD, even those diagnosed with mild AD [10].

A literature review of the linguistic features of connected speech in patients with AD reported that AD is associated with the following characteristics: reduced speech speed and spontaneity, including increased repetition and correction; simplified syntax and sentence structure, including shorter and more grammatically simple sentences; word-finding difficulties and increased use of pronouns; noun and verb inflection errors; and reduced semantic content of speech and non-informative utterances with low density and efficiency of ideas [11,12,13]. Satt et al. [14] analyzed the speech characteristics of people with AD using the repetitive syllables “pa-ta-ka”. They reported that the speech was more irregular than that of healthy elderly people [14]. König et al. [15] performed automatic speech analysis of healthy control participants and individuals with MCI. The results revealed high classification accuracy, indicating the usefulness of the evaluation. In another study, König et al. [16] reported the potential value of using speech analysis and mobile applications for accurate automatic differentiation of MCI and AD. These studies have prompted researchers to apply intelligent algorithms to analyze speech features, with findings suggesting the possibility of analyzing independent biomarkers of spontaneous speech and emotional responses using artificial neural networks as features of spontaneous speech and emotional responses [17,18].

Artificial intelligence (AI) has been used in research to differentiate AD from other conditions or typical aging. Previous studies have reported that AI can identify patients diagnosed with AD on the basis of genes, MRI images, and electronic health record data [19]. AI technology is now also beginning to be applied to speech analysis. Fristed et al. [20] successfully predicted MCI/mild AD using an AI system with a smartphone, and Agbavor and Liang [21] successfully distinguished AD from non-AD by analyzing the voices of elderly individuals. In recent years, there has been an increase in research using AI systems to try to diagnose AD at an early stage by analyzing the patient’s speech [22]. Although a scoping review of research on this topic has been published [23,24], it has been pointed out that caution is still warranted regarding the accuracy of AI-based AD diagnosis [25].

The phonetic properties of AD speech need to be analyzed in a language- and country-specific way. Phonetic analyses of the speech of patients with AD have been undertaken in English and Bengali [26], but such investigations have not yet been undertaken in Japanese. Thus, in the current study, we compared the speech characteristics of Japanese patients with AD with the speech of healthy elderly people in Japan.

## 2. Methods

### 2.1. Design and Participants

A case-control design was used to compare two groups: patients with AD and healthy age-matched elderly participants. Patients with AD were recruited from one hospital in the Hokuriku region of Japan. Healthy elderly participants were recruited from a private nursing home in the Tokai region of Japan. Regarding the selection method for the subjects, AD patients were all cases at the relevant hospital, and healthy elderly people were consecutive cases who consented to the study at the nursing home. The exclusion criteria for subjects were the presence of aphasia and difficulty obtaining consent from the patient. We did not apply selection criteria regarding age or gender.

### 2.2. Voice Recording and Cognitive Assessment

Participants’ speech was recorded for approximately 20 min, comprising 3 min of free conversation and 15 min of cognitive function assessment, followed by a further 3 min of free conversation. The aim of the 3-min free conversation before and after the cognitive assessment was to relieve the participant’s tension and to record the conversation as a sentence, not just at the word level. The recording environment was a standard examination room or meeting room, not a soundproof room specially prepared for recording. A standard webcam with a microphone was placed between the examiner (an occupational therapist or a person trained in cognitive function testing for this study) and the participant, at a distance of 40 cm from both. In this environment, various types of noise and interference may affect data, including air conditioning noise from air conditioners and ventilation fans, fan noise from personal computers, conversations, chatter, noise outside the testing room or meeting room, and the noise of writing, putting down a pen, and turning over paper during a test.

Audio files were saved in the RIFF WAV format with a sampling rate of 44.1 kHz. These audio files, including all the responses from the Mini-Mental State Examination (MMSE) and Hasegawa’s Dementia Scale-Revised (HDS-R) and the free conversation before and after the questions, were manually segmented. We used MMSE and HDS-R for speech recording evaluation because we sought to prioritize simplicity and avoid subject fatigue over a long period of time, rather than for the purpose of diagnosing AD. The MMSE and HDS-R were administered face-to-face, and they were also administered consecutively. At this time, we excluded any noise or interference that could be clearly distinguished by the human ear from the data to be extracted. In addition, to account for the fact that the data may have included noise that cannot be distinguished by the human ear, we extracted the “noise only” parts from the original audio data that did not include the voices of the test subjects, mixed up all of this noise, then synthesized it into all of the audio files after extraction to reduce the impact of noise between the test subjects. The MMSE and HDS-R were used to assess cognitive function. The MMSE covers several domains, including orientation, memory, attention, numeracy, verbal ability, and visuospatial cognition [27]. The maximum score is 30, and a score between 24 and 27 is considered indicative of MCI [28]. The HDS-R, like the MMSE, is a 30-point test that measures cognitive function in elderly people. It consists of nine questions regarding age, date, location, word recall, subtraction, number recall, word recall, immediate recall of objects, and enumeration of vegetable names. The HDS-R cutoff score was set at 20 [29].

### 2.3. Speech Feature Extraction

We investigated differences between the healthy elderly individuals (Healthy Elderly) and AD patients (Patients) in the speech features computed from all responses on the MMSE and the HDS-R assessments, as well as the free conversation before and after the questions. First, we performed preprocessing to extract the speech segments to be analyzed. The procedure was as follows: (1) We combined the free conversation before, during, and after the test as a single audio file and extracted only the subject’s speech segments. (2) We created blocks of speech by shifting 10-s blocks of speech by 5 s. Finally, (3) from the created speech blocks, we extracted features such as the average intensity. 

Regarding the methods for examining voice features, we selected speech features with reference to the open-source toolkit openSMILE [30], used by Maikusa et al. [31] and Toyama et al. [32], and the open-source software tool Praat (version 6.4.02) [33], used by Shimoda et al. [34]. All of these tools are widely used internationally for speech analysis, and the speech features and basic statistics output by these tools were also used as references in this study. These measures included the spectrum, intensity, and fundamental frequency. In addition to these measures, we also investigated the minimum variation for intensity and fundamental frequency.

(1) Spectrum: This is a basic indicator for analyzing speech features and can also be confirmed visually as a waveform. We sampled the shapes of the spectra of AD patients and healthy people in advance and speculated that they might contain important information for distinguishing between healthy people and AD patients. We confirmed that the AD patients exhibited a gentle downward slope, whereas the healthy people tended to exhibit a rapid downward slope around 6,000 Hz. Therefore, we calculated a total of seven features (3 × 2 + 1), including the center of gravity, skewness, kurtosis, and standard deviation of the spectrum.

(2) Intensity and fundamental frequency: In addition to intensity, which expresses the strength of the voice, the fundamental frequency determines the pitch (the height of the sound) that expresses gender and individual differences. These were selected as features for investigating the impact of AD on patients’ speech. Thus, the mean, median, minimum, maximum, 15th percentile, 85th percentile, standard deviation, skewness, and kurtosis were examined, giving a total of 18 (2 × 9) features.

(3) Minimum change in intensity and fundamental frequency (∆): To capture slight variations, we examined the data to detect minute changes in these measures. There was a total of 18 (2 × 9) features: mean, median, minimum value, maximum value, 15th percentile, 85th percentile, standard deviation, skewness, and kurtosis.

Therefore, the overall size of the feature set was 43 (7 + 18 + 18) features. These vocal features were computed with LibROSA (v. 0.10.1) in Python (v. 3.8.0) and statistically processed using NumPy (v. 1.24.3) and SciPy (v. 1.10.1). To compare the patient group with the healthy elderly group, a *t*-test was used at the 5% significance level.

### 2.4. Classifiers and Performance Evaluation

We evaluated the performance of three classifiers—logistic regression (LR) [35], support vector machine (SVM) [36], and random forest (RF) [37]—on the speech features that differed between the Healthy Elderly and Patient groups. 

(1) Logistic regression (LR): As a method used in classification problems, LR predicts the category (e.g., success/failure) of a binary variable from input data. The model performs a linear combination of features and then uses a sigmoid function to convert the results into a probability between 0 and 1. Using these probabilities, LR predicts whether the data belong to a particular class. LR is simple and easy to interpret, but because of its linearity assumption, other methods may be needed for complex data patterns. This method is considered to be one of the leading candidates among the classifiers used in this study.

(2) Support vector machine (SVM): Like LR, SVM is a method used for classification problems, and it finds the optimal boundary (hyperplane) for dividing the data. SVM maximizes the margin between classes and utilizes data points near the boundary, called support vectors. Because SVM can also handle non-linear data, we selected it for comparison with LR.

(3) Random forest (RF): This method combines multiple decision trees to make predictions. Each decision tree is trained using a random subset of the data, and a random subset of the features is selected for the split. In this way, a variety of models are generated to improve the stability and accuracy of the predictions. As this classifier combines multiple models, it was selected in the hope that it would produce higher accuracy than LR and SVM.

When using LR, SVM, or RF, it is typically necessary to consider the fine-tuning of several parameters. However, in this case, we used a single explanatory variable to observe factors that may affect AD prediction and did not perform parameter optimization using grid search. As such, this verification process is a task that should be carried out before building a full-fledged AI model. However, in the current study, we carried out this step because AD prediction is a meaningful and important initiative. This performance evaluation was conducted by generating models using a single speech feature as a single explanatory variable, without combining the speech features. Although we do not believe that an AI model can be created with only a single feature, the primary purpose of the current study was to get a feel for which features might have an impact on AD prediction. All the models were computed using the scikit-learn library (v. 1.3.0) in Python (v. 3.8.0). The speech datasets used were fivefold cross-validated, with 80% of the data in each dataset used for training and parameter tuning and the remaining 20% used for performance evaluation.

### 2.5. Ethical Procedures

Ethics approval was obtained from the Ethical Review Committee of Niigata University of Health and Welfare (Ref: 18762-211126). All the participants gave informed written consent to take part in this study.

## 3. Results

Eighty-three patients with AD and 75 healthy elderly participants with complete data were included in the analysis. Participants’ characteristics are shown in Table 1. There were no differences in age and gender between the two groups. The mean MMSE scores for the Patients and Healthy Elderly groups were 15.9 ± 0.7 and 28.1 ± 0.3, respectively (*p* < 0.001). Mean scores for the HDS-R were 12.0 ± 0.8 and 26.5 ± 0.7 for the Patient and Healthy Elderly groups, respectively (*p* < 0.001).

Table 2 shows the results of the speech feature analysis. The spectrum was significantly different between the two groups for all the components. For intensity, significant differences were found between the two groups for all the components except for the standard deviation. There were no differences in the mean values and 15th and 85th percentiles of the fundamental frequency. In addition, there were no differences in the mean, standard deviation, or skewness of the infinitesimal changes (∆) of the fundamental frequency, although there were some differences in the infinitesimal changes of the intensity (∆) (except for the mean).

Table 3, Table 4, Table 5, Table 6 and Table 7 show the performance evaluation results according to the three classifiers (i.e., LR, SVM, and RF). The performance evaluation was conducted on the voice features that showed significant differences in Table 2. The mean and standard deviation of the center of gravity, skewness, and kurtosis of the spectrum had high values for accuracy, F1 score, and area under the curve (AUC). In particular, the mean AUC values of the fivefold cross-validation of the standard deviation of the kurtosis of the spectrum were 0.977 ± 0.012 (LR), 0.971 ± 0.015 (SVM), and 0.952 ± 0.016 (RF), indicating that the classifier with either LR or SVM had the highest performance. Figure 1 shows the receiver operating characteristic curves from the fivefold cross-validation for Kurtosis_SD, and all graphs show the AUC. The X-axis (false positive rate) shows the percentage of healthy people whose speech characteristics were incorrectly judged as those of AD patients. The Y-axis (true positive rate) shows the percentage of AD patients whose speech characteristics were correctly judged as those of AD patients. Thus, the performance of this analysis yielded a low false positive rate and an extremely high true positive rate.

## 4. Discussion

This study compared the speech characteristics of patients with AD and those of healthy elderly people for machine learning. The extracted vocal characteristics were compared in terms of the spectrum, intensity, and fundamental frequency, and the mean value of infinitesimal changes (∆) in the intensity and fundamental frequency, skewness, kurtosis, and other factors. The results showed that most of the factors differed between the two groups and that the mean and standard deviation of the center of gravity, skewness, and kurtosis of the spectrum performed most highly in the performance evaluation using the three classifiers (i.e., LR, SVM, and RF).

Regarding the voice characteristics of patients with AD, Meilán et al. [38] and Martínez-Nicolás et al. [39] reported that variations in the percentage of voice breaks, number of periods of voice, number of voice breaks, shimmer (amplitude perturbation quotient), and noise-to-harmonics ratio were promising for discriminating people with AD from typically aging individuals. Shimoda et al. [34] examined a predictive model for AD by extracting intensity and pitch, as well as center of gravity, skewness, kurtosis, and standard deviation. In our study, we focused on the spectrum, intensity, and fundamental frequency and investigated their characteristics in detail. The results showed excellent discriminative ability, providing new evidence to advance the field.

König et al. [15,16] reported that AD can be distinguished by temporal acoustic features, such as the length of voiced (periodic) segments and silent segments. Themistocleous et al. [40] also observed that acoustic features, such as vowel duration, vowel formant frequency, and fundamental frequency, are effective predictors of AD. Furthermore, articulatory agility is reduced in AD [41], and the incidence of apraxia of motion is higher [42], suggesting that the effects of cognitive decline in AD are particularly evident in the speech spectrum and that the detailed data regarding kurtosis and skewness are helpful for characterizing AD. The clear discrimination between the individuals with AD and typical elderly people in the SD of kurtosis and skewness of the spectrum suggests that it is difficult for people with AD to maintain the constant distribution of energy required for speech production. However, the mechanism of this change was unclear, and further investigation is needed.

The current findings have the potential to provide a non-invasive assessment tool for the initial diagnosis of AD. In the future, we intend to collect more case data and advance machine learning to develop an AI that can discriminate between individuals with AD and healthy elderly people.

The current study involved several limitations. First, the audio files were not recorded in a special environment such as a soundproof room. As explained in the Methods section, there was noise from the air conditioner, ventilation fan, and personal computer fan, as well as noise from outside the meeting room. The authors listened to these noises and excluded them from the analysis. However, because this process was entirely manual, errors may have occurred. For this reason, we cannot rule out the possibility that noise affected the analysis results. The recordings of AD patients were obtained in a hospital environment, whereas the recordings of healthy elderly people were obtained in a different environment. This difference in recording environments and the influence of environmental sounds may have impacted the results. However, because the recordings were made in multiple rooms in both the hospital and the residential facility, we believe that these differences are unlikely to have caused substantial bias in the results between the two groups. Second, the recording site for individuals with AD and the facility in which the healthy elderly people were assessed were located in different regions of Japan. Thus, the possibility that dialect had an effect on findings cannot be ruled out. We plan to investigate this issue in future studies by recording data in other regions. Third, this study did not control for age, gender, and severity of AD. Additionally, the sample size was relatively small. In future research, we plan to increase the number of samples, adjust for age and gender, and analyze the data in a way that allows us to classify the speech of AD patients and healthy elderly people while eliminating the effects of confounding factors such as the severity of AD.

## 5. Conclusions

In this study, we focused on changes in the speech of patients with AD and investigated the characteristics of those differences through comparison with healthy elderly people using machine learning. Significant differences were found between the two groups in all components of the spectrum, including the mean value of the center of gravity, skewness, kurtosis, and standard deviation. High AUCs were obtained for the center of gravity, mean skewness, kurtosis, and standard deviation of the spectrum in the performance evaluation. These speech features of patients with AD obtained by machine learning pave the way for early non-invasive AD diagnosis in the future.

## Figures and Tables

**Figure 1 healthcare-12-02194-f001:**
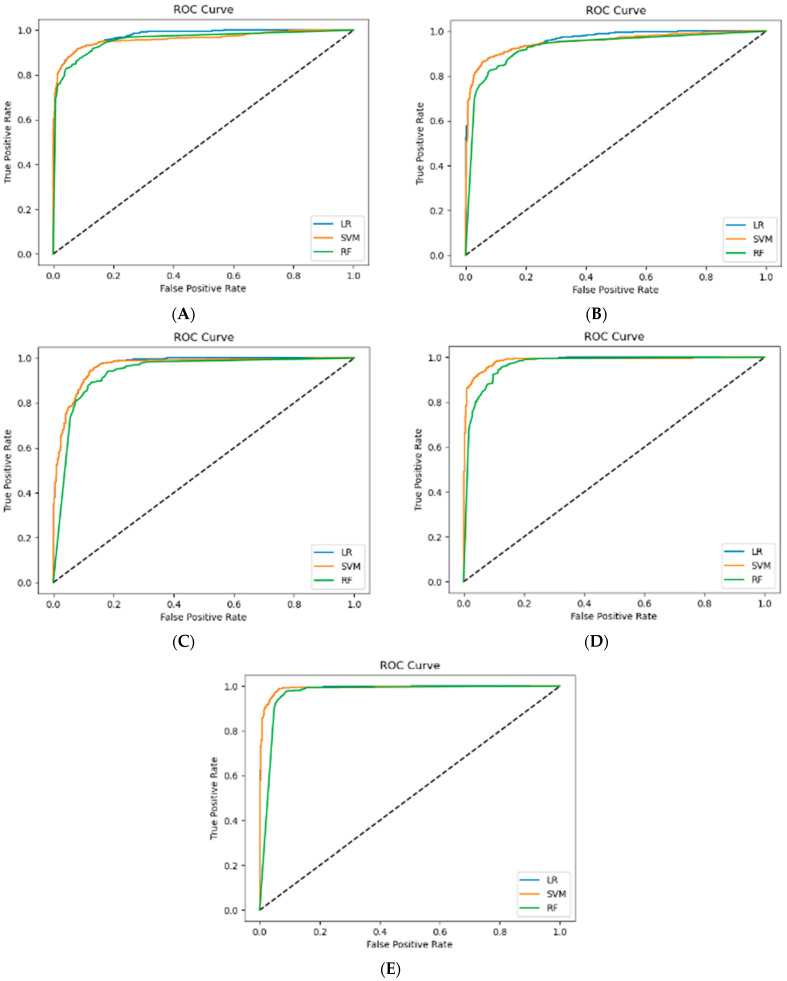
ROC curves for performance evaluation of spectrum: Kurtosis_SD by three classifiers (LR, logistic regression; SVM, support vector machine; and RF, random forest). The X-axis (false positive rate) shows the percentage of healthy people whose voice characteristics were mistakenly judged to be those of AD patients. The Y-axis (true positive rate) shows the percentage of AD patients whose voice characteristics were correctly judged to be those of AD patients. (**A**–**E**) in the graph indicate that it is divided into five parts.

**Table 1 healthcare-12-02194-t001:** Participants’ characteristics.

Characteristics	Patients, n = 83	Healthy Elderly, n = 75	*p*-Value
Age (years)	81.6 ± 0.8	80.4 ± 0.7	0.864
Gender (M/F)	33/50	28/47	0.097
MMSE	15.9 ± 0.7	28.1 ± 0.3	<0.001
HDS-R	12.0 ± 0.8	26.5 ± 0.7	<0.001

Abbreviations: MMSE, Mini-Mental State Examination; HDS-R, Hasegawa’s Dementia Scale-Revised.

**Table 2 healthcare-12-02194-t002:** Speech features.

Features	Patients	Healthy Elderly	*p*-Value
**Spectrum:**			
Center of gravity_mean	1230.873 ± 182.685	1546.085 ± 211.472	<0.001
Center of gravity_SD	0.623 ± 0.187	0.340 ± 0.120	<0.001
Skewness_mean	−0.516 ± 0.405	−0.991 ± 0.147	<0.001
Skewness_SD	566.606 ± 136.623	674.966 ± 141.166	<0.001
Kurtosis_mean	0.354 ± 0.106	0.202 ± 0.056	<0.001
Kurtosis_SD	0.665 ± 0.449	0.187 ± 0.081	<0.001
Standard deviation	1.431 ± 0.752	1.388 ± 0.636	<0.05
**Intensity:**			
Mean	0.040 ± 0.022	0.039 ± 0.019	<0.001
Median	0.033 ± 0.021	0.031 ± 0.019	<0.001
Minimum	0.002 ± 0.001	0.003 ± 0.001	<0.001
Maximum	0.181 ± 0.091	0.173 ± 0.077	<0.001
15th percentile	0.002 ± 0.001	0.003 ± 0.001	<0.001
85th percentile	0.003 ± 0.002	0.003 ± 0.001	<0.001
Standard deviation	0.032 ± 0.017	0.032 ± 0.014	0.559
Skewness	1.395 ± 0.721	1.288 ± 0.639	<0.001
Kurtosis	2.968 ± 4.298	2.261 ± 3.402	<0.001
**Intensity delta:**			
Mean	0.000003 ± 0.00005	0.000003 ± 0.00005	0.982
Median	−0.0003 ± 0.0002	−0.0002 ± 0.0002	<0.001
Minimum	−0.037 ± 0.022	−0.034 ± 0.018	<0.001
Maximum	0.041 ± 0.023	0.038 ± 0.019	<0.001
15th percentile	−0.030 ± 0.017	−0.028 ± 0.014	<0.001
85th percentile	−0.020 ± 0.011	−0.019 ± 0.009	<0.001
Standard deviation	0.007 ± 0.004	0.007 ± 0.003	<0.001
Skewness	0.538 ± 0.610	0.504 ± 0.531	<0.05
Kurtosis	7.779 ± 7.947	6.830 ± 6.816	<0.001
**Fundamental frequency:**			
Mean	87.747 ± 19.786	88.441 ± 17.607	0.118
Median	84.184 ± 19.467	85.698 ± 17.536	<0.001
Minimum	55.374 ± 9.375	54.776 ± 8.511	<0.01
Maximum	153.802 ± 45.689	148.902 ± 41.004	<0.001
15th percentile	56.071 ± 9.853	55.675 ± 9.049	0.077
85th percentile	58.237 ± 11.260	58.119 ± 10.181	0.642
Standard deviation	18.760 ± 9.557	18.092 ± 7.731	<0.01
Skewness	1.054 ± 0.968	0.753 ± 0.785	<0.001
Kurtosis	2.290 ± 3.997	1.205 ± 2.622	<0.001
**Fundamental frequency delta:**			
Mean	−0.006 ± 0.076	−0.009 ± 0.063	0.182
Median	−0.136 ± 0.210	−0.103 ± 0.181	<0.001
Minimum	−49.892 ± 33.383	−43.160 ± 27.234	<0.001
Maximum	61.145 ± 32.171	55.616 ± 26.548	<0.001
15th percentile	−34.962 ± 22.897	−31.435 ± 16.796	<0.001
85th percentile	−15.236 ± 6.793	−16.305 ± 5.932	<0.001
Standard deviation	6.025 ± 2.536	5.969 ± 2.001	0.308
Skewness	1.952 ± 2.737	1.900 ± 2.205	0.380
Kurtosis	51.307 ± 36.997	38.432 ± 27.797	<0.001

**Table 3 healthcare-12-02194-t003:** Classifiers and performance evaluation: Spectrum.

Classifiers	Accuracy	F1 Score	AUC
**Spectrum: Center of gravity_mean**			
LR	0.794 ± 0.048	0.765 ± 0.060	0.874 ± 0.046
SVM	0.796 ± 0.047	0.760 ± 0.061	0.844 ± 0.050
RF	0.731 ± 0.032	0.698 ± 0.039	0.821 ± 0.043
**Spectrum: Center of gravity_SD**			
LR	0.837 ± 0.032	0.812 ± 0.036	0.908 ± 0.036
SVM	0.838 ± 0.032	0.813 ± 0.035	0.889 ± 0.026
RF	0.760 ± 0.033	0.733 ± 0.031	0.850 ± 0.035
**Spectrum: Skewness_mean**			
LR	0.838 ± 0.012	0.806 ± 0.013	0.904 ± 0.023
SVM	0.838 ± 0.011	0.802 ± 0.014	0.878 ± 0.015
RF	0.762 ± 0.024	0.734 ± 0.024	0.850 ± 0.023
**Spectrum: Skewness_SD**			
LR	0.683 ± 0.019	0.632 ± 0.026	0.723 ± 0.020
SVM	0.683 ± 0.018	0.615 ± 0.031	0.705 ± 0.021
RF	0.594 ± 0.010	0.544 ± 0.020	0.636 ± 0.019
**Spectrum: Kurtosis_mean**			
LR	0.840 ± 0.035	0.815 ± 0.039	0.932 ± 0.023
SVM	0.843 ± 0.037	0.823 ± 0.039	0.911 ± 0.029
RF	0.792 ± 0.027	0.768 ± 0.024	0.890 ± 0.026
**Spectrum: Kurtosis_SD**			
LR	0.920 ± 0.021	0.909 ± 0.020	0.977 ± 0.012
SVM	0.920 ± 0.022	0.910 ± 0.021	0.971 ± 0.015
RF	0.888 ± 0.023	0.876 ± 0.022	0.952 ± 0.016
**Spectrum: Standard deviation**			
LR	0.556 ± 0.020	0.048 ± 0.076	0.462 ± 0.055
SVM	0.558 ± 0.037	0.255 ± 0.084	0.545 ± 0.047
RF	0.514 ± 0.012	0.455 ± 0.010	0.510 ± 0.013

Abbreviations: AUC, area under the curve; LR, logistic regression; SD, standard deviation; SVM, support vector machine; RF, random forest.

**Table 4 healthcare-12-02194-t004:** Classifiers and performance evaluation: Intensity.

Classifiers	Accuracy	F1 Score	AUC
**Intensity: Mean**			
LR	0.554 ± 0.018	0.001 ± 0.003	0.501 ± 0.063
SVM	0.552 ± 0.032	0.236 ± 0.076	0.537 ± 0.043
RF	0.524 ± 0.010	0.469 ± 0.011	0.518 ± 0.015
**Intensity: Median**			
LR	0.554 ± 0.018	0.001 ± 0.003	0.508 ± 0.053
SVM	0.548 ± 0.018	0.133 ± 0.113	0.509 ± 0.027
RF	0.510 ± 0.015	0.452 ± 0.012	0.510 ± 0.021
**Intensity: Minimum**			
LR	0.553 ± 0.018	0.000 ± 0.000	0.685 ± 0.090
SVM	0.697 ± 0.059	0.530 ± 0.114	0.691 ± 0.070
RF	0.605 ± 0.025	0.546 ± 0.046	0.652 ± 0.054
**Intensity: Maximum**			
LR	0.554 ± 0.018	0.098 ± 0.097	0.463 ± 0.062
SVM	0.573 ± 0.023	0.284 ± 0.064	0.537 ± 0.034
RF	0.519 ± 0.011	0.462 ± 0.024	0.512 ± 0.012
**Intensity: 15th percentile**			
LR	0.553 ± 0.018	0.000 ± 0.000	0.674 ± 0.091
SVM	0.697 ± 0.062	0.522 ± 0.118	0.682 ± 0.072
RF	0.598 ± 0.031	0.545 ± 0.045	0.649 ± 0.058
**Intensity: 85th percentile**			
LR	0.553 ± 0.018	0.000 ± 0.000	0.634 ± 0.086
SVM	0.668 ± 0.041	0.486 ± 0.098	0.665 ± 0.054
RF	0.605 ± 0.024	0.558 ± 0.033	0.655 ± 0.039
**Intensity: Skewness**			
LR	0.554 ± 0.023	0.155 ± 0.053	0.542 ± 0.071
SVM	0.554 ± 0.028	0.252 ± 0.076	0.525 ± 0.041
RF	0.512 ± 0.008	0.455 ± 0.019	0.506 ± 0.005
**Intensity: Kurtosis**			
LR	0.559 ± 0.022	0.172 ± 0.034	0.556 ± 0.064
SVM	0.563 ± 0.022	0.284 ± 0.040	0.541 ± 0.038
RF	0.516 ± 0.012	0.456 ± 0.015	0.513 ± 0.012

**Table 5 healthcare-12-02194-t005:** Classifiers and performance evaluation: Intensity delta.

Classifiers	Accuracy	F1 Score	AUC
**Intensity delta: Median**			
LR	0.553 ± 0.018	0.000 ± 0.000	0.518 ± 0.036
SVM	0.554 ± 0.018	0.128 ± 0.091	0.522 ± 0.020
RF	0.517 ± 0.003	0.427 ± 0.011	0.501 ± 0.013
**Intensity delta: Minimum**			
LR	0.555 ± 0.018	0.011 ± 0.017	0.518 ± 0.069
SVM	0.561 ± 0.014	0.230 ± 0.045	0.529 ± 0.015
RF	0.504 ± 0.008	0.445 ± 0.017	0.507 ± 0.013
**Intensity delta: Maximum**			
LR	0.555 ± 0.019	0.018 ± 0.023	0.528 ± 0.065
SVM	0.559 ± 0.025	0.256 ± 0.044	0.527 ± 0.013
RF	0.513 ± 0.011	0.458 ± 0.015	0.521 ± 0.018
**Intensity delta: 15th percentile**			
LR	0.553 ± 0.018	0.001 ± 0.003	0.519 ± 0.070
SVM	0.570 ± 0.018	0.269 ± 0.044	0.539 ± 0.034
RF	0.507 ± 0.014	0.452 ± 0.013	0.502 ± 0.014
**Intensity delta: 85th percentile**			
LR	0.553 ± 0.018	0.000 ± 0.000	0.513 ± 0.072
SVM	0.569 ± 0.020	0.264 ± 0.051	0.545 ± 0.032
RF	0.499 ± 0.014	0.437 ± 0.009	0.493 ± 0.013
**Intensity delta: Standard deviation**			
LR	0.553 ± 0.018	0.000 ± 0.000	0.477 ± 0.058
SVM	0.565 ± 0.023	0.276 ± 0.053	0.539 ± 0.028
RF	0.505 ± 0.014	0.445 ± 0.023	0.499 ± 0.013
**Intensity delta: Skewness**			
**LR**	0.554 ± 0.018	0.027 ± 0.033	0.510 ± 0.043
SVM	0.564 ± 0.018	0.260 ± 0.047	0.552 ± 0.029
RF	0.512 ± 0.013	0.453 ± 0.021	0.505 ± 0.013
**Intensity delta: Kurtosis**			
LR	0.556 ± 0.020	0.110 ± 0.030	0.542 ± 0.056
SVM	0.553 ± 0.025	0.203 ± 0.053	0.528 ± 0.047
RF	0.511 ± 0.008	0.446 ± 0.011	0.507 ± 0.005

**Table 6 healthcare-12-02194-t006:** Classifiers and performance evaluation: Fundamental frequency.

Classifiers	Accuracy	F1 Score	AUC
**Fundamental frequency: Median**			
LR	0.548 ± 0.022	0.080 ± 0.093	0.380 ± 0.053
SVM	0.555 ± 0.046	0.176 ± 0.101	0.499 ± 0.059
RF	0.552 ± 0.026	0.392 ± 0.029	0.540 ± 0.028
**Fundamental frequency: Minimum**			
LR	0.539 ± 0.033	0.043 ± 0.054	0.455 ± 0.105
SVM	0.525 ± 0.051	0.114 ± 0.073	0.473 ± 0.086
RF	0.488 ± 0.035	0.297 ± 0.158	0.514 ± 0.005
**Fundamental frequency: Maximum**			
LR	0.540 ± 0.030	0.086 ± 0.101	0.543 ± 0.106
SVM	0.578 ± 0.060	0.413 ± 0.067	0.583 ± 0.053
RF	0.557 ± 0.033	0.440 ± 0.040	0.559 ± 0.029
**Fundamental frequency: Standard deviation**			
LR	0.548 ± 0.023	0.055 ± 0.054	0.510 ± 0.065
SVM	0.565 ± 0.031	0.293 ± 0.019	0.557 ± 0.030
RF	0.519 ± 0.009	0.462 ± 0.022	0.521 ± 0.014
**Fundamental frequency: Skewness**			
LR	0.590 ± 0.016	0.378 ± 0.041	0.587 ± 0.017
SVM	0.588 ± 0.016	0.352 ± 0.053	0.584 ± 0.016
RF	0.515 ± 0.007	0.456 ± 0.013	0.521 ± 0.007
**Fundamental frequency: Kurtosis**			
LR	0.582 ± 0.015	0.302 ± 0.026	0.594 ± 0.018
SVM	0.585 ± 0.012	0.365 ± 0.029	0.582 ± 0.017
RF	0.516 ± 0.004	0.460 ± 0.019	0.517 ± 0.003

**Table 7 healthcare-12-02194-t007:** Classifiers and performance evaluation: Fundamental frequency delta.

Classifiers	Accuracy	F1 Score	AUC
**Fundamental frequency delta: Median**			
LR	0.542 ± 0.037	0.235 ± 0.129	0.538 ± 0.061
SVM	0.565 ± 0.046	0.192 ± 0.084	0.528 ± 0.053
RF	0.567 ± 0.032	0.244 ± 0.080	0.521 ± 0.022
**Fundamental frequency delta: Minimum**			
LR	0.587 ± 0.027	0.316 ± 0.043	0.546 ± 0.064
SVM	0.591 ± 0.022	0.343 ± 0.061	0.571 ± 0.028
RF	0.530 ± 0.009	0.475 ± 0.020	0.545 ± 0.013
**Fundamental frequency delta: Maximum**			
LR	0.578 ± 0.016	0.250 ± 0.047	0.546 ± 0.058
SVM	0.600 ± 0.020	0.410 ± 0.048	0.594 ± 0.028
RF	0.516 ± 0.013	0.454 ± 0.010	0.532 ± 0.011
**Fundamental frequency delta: 15th percentile**			
LR	0.571 ± 0.021	0.241 ± 0.029	0.518 ± 0.059
SVM	0.581 ± 0.020	0.320 ± 0.044	0.566 ± 0.011
RF	0.524 ± 0.015	0.467 ± 0.031	0.535 ± 0.016
**Fundamental frequency delta: 85th percentile**			
LR	0.561 ± 0.029	0.112 ± 0.107	0.572 ± 0.062
SVM	0.588 ± 0.028	0.390 ± 0.051	0.579 ± 0.039
RF	0.528 ± 0.011	0.472 ± 0.016	0.540 ± 0.013
**Fundamental frequency delta: Kurtosis**			
LR	0.608 ± 0.017	0.400 ± 0.042	0.619 ± 0.029
SVM	0.612 ± 0.019	0.437 ± 0.046	0.604 ± 0.028
RF	0.514 ± 0.008	0.457 ± 0.011	0.523 ± 0.007

## Data Availability

Anonymized data will be shared upon reasonable request from qualified academic investigators for the sole purpose of replicating procedures and results presented in the article.

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
