# Peer review of "Analysis of Speech Features in Alzheimer’s Disease with Machine Learning: A Case-Control Study"

_healthcare, 2024, doi:10.3390/healthcare12212194_

Round 1
Reviewer 1 Report
Comments and Suggestions for Authors
Dear Authors,
Thank you for submitting your manuscript titled "Analysis of Speech Features in Alzheimer’s Disease for Machine Learning: A Case-Control Study" which explores the use of speech analysis to differentiate between individuals with Alzheimer’s disease (AD) and healthy elderly participants. The study involves recording speech during cognitive assessments, such as the Mini-Mental State Examination (MMSE) and Hasegawa’s Dementia Scale-Revised (HDS-R), as well as free conversations. The authors extracted multiple speech features, including spectrum, intensity, and fundamental frequency. They applied machine learning classifiers (logistic regression, support vector machine, and random forest) to evaluate the performance of these features in distinguishing AD patients from healthy controls. Significant differences were found in various speech features, particularly in the spectrum analysis, which suggests that speech characteristics can serve as potential non-invasive markers for early AD diagnosis. The study highlights the potential of machine learning in analyzing speech for detecting neurodegenerative diseases.
The manuscript presents a well-structured and relevant study on the application of machine learning to speech features for Alzheimer’s disease detection. The topic is timely, given the increasing interest in non-invasive diagnostic methods for neurodegenerative diseases. The research design is appropriate, and the methods are sound, particularly the use of multiple classifiers to validate the performance of speech features. However, the overall quality could be significantly improved by addressing several areas:
- Methodological Transparency: The justification for selecting specific speech features and classifiers could be clearer. Additionally, more details are needed regarding the cross-validation process and how potential biases (e.g., dialectal variations, non-standard recording environments) were managed.
- Results Presentation: The results are valuable but could be presented in a more accessible way. Figures and tables need clearer labelling and better explanation to enhance readability and comprehension.
- Discussion and Limitations: The discussion is relevant but would benefit from a more detailed exploration of the limitations, particularly the potential influence of non-controlled variables like dialects and recording conditions.
- Language and Fluency: The quality of the English language is generally acceptable, though moderate editing is required for clarity and smoother transitions between sections.
Here are my specific comments in detail;
Introduction: The introduction offers a good summary of the relevance of speech analysis for Alzheimer’s disease (AD) detection but could be enriched by integrating more recent studies on non-invasive screening methods using AI in neurodegenerative diseases. Some key references are missing, such as recent advancements in speech-based diagnostic tools for AD.
Methods: The case-control design is appropriate, but there should be a clearer explanation of the criteria used for participant selection. The justification for not controlling for dialectal variations and the non-standard recording environment could be discussed in greater detail. The speech feature extraction method is well-described, but additional detail is needed on why certain features were chosen for the analysis. Also, there is limited information on the statistical power of the classifiers used. Furthermore, the section could benefit from a more comprehensive explanation of the cross-validation process and model performance metrics.
Results: The results are generally well presented, but some figures and tables are difficult to interpret at first glance. Specifically, Figure 1 (ROC curves) would benefit from clearer axis labeling, and the discussion of the AUC values could be expanded to explain their practical significance more fully. Moreover, certain aspects of the classifiers' performance (e.g., Random Forest) require further elaboration.
Discussion: The conclusions are generally supported by the data, but the limitations related to recording environments and the relatively small sample size should be emphasized more thoroughly. A stronger link between the findings and the potential for future clinical application would strengthen the discussion.
The language is understandable, but there are several instances where phrasing could be improved for clarity and fluency. There are minor grammatical issues, and some sections are overly technical, which could make it difficult for a broader audience to grasp the significance of the findings. Additionally, transitions between sections can be smoother.
My final Comments for Authors:
- Introduction: The background is well-established, but the inclusion of more recent studies (within the past two years) would strengthen the relevance of the study.
- Methods: The choice of classifiers and features could be better justified, and more details on cross-validation and statistical power are necessary. Additionally, the limitations of the non-standard recording environment should be discussed in more depth.
- Results: Some tables and figures are difficult to interpret. I suggest improving the labeling and providing more context for the AUC values to help readers understand the clinical significance of your findings.
- Discussion: Expand the limitations section to cover all potential sources of bias, including the differences in recording environments and dialectal influences. Also, more emphasis on the practical implications of these findings for clinical practice would enhance the conclusion.
- English Language: The paper would benefit from moderate editing to improve clarity and fluency, especially in the transitions between sections and the technical descriptions.
Thank you for your valuable contribution to this important area of research. I look forward to your revisions and the final version of your manuscript.
Sincerely,
Comments on the Quality of English LanguageThe language is understandable, but there are several instances where phrasing could be improved for clarity and fluency. There are minor grammatical issues, and some sections are overly technical, which could make it difficult for a broader audience to grasp the significance of the findings. Additionally, transitions between sections can be smoother.
Author Response
We would like to express our sincere gratitude for your kind and appropriate advice. We would also like to thank you for your comments and have attached a file with our response to them. Please check it out.

Reviewer 2 Report
Comments and Suggestions for Authors
The study presents valuable insights into the speech characteristics of Alzheimer's disease patients. However, it would benefit from a more detailed description of the methods used for data collection and analysis. Additionally, including more recent references could strengthen the background and context of the research. Consider discussing the implications of your findings in greater depth, particularly how they might influence future research or clinical practice.
- Participant Selection: Provide more information on the criteria for selecting participants, including any inclusion or exclusion criteria.
- Recording Environment: Describe the specific conditions under which the recordings were made, including any potential sources of noise or distraction that could affect the data.
- Speech Analysis Techniques: Elaborate on the specific machine learning algorithms or statistical methods used for analyzing the speech features. Include details on how features were extracted and any preprocessing steps taken.
- Cognitive Assessment Tools: Specify the cognitive assessment tools used (e.g., Mini-Mental State Examination) and how they were administered.
- Data Handling: Explain how the data was managed, including any software used for analysis and how the results were validated.
The overall quality of English is moderate, with some areas requiring editing for clarity and coherence. Specific phrases may be difficult to understand, and restructuring certain sentences could enhance readability. It is recommended to have a native speaker or professional editor review the manuscript to improve the flow and precision of the language used.
-
Line 90-92: "Participants’ speech was recorded for approximately 20 minutes, comprising 3 minutes of free conversation, 15 minutes of cognitive function assessment, followed by a further 3 minutes of free conversation."
- Suggestion: Clarify the purpose of each segment of the recording and how it contributes to the overall analysis.
-
Line 240-243: "This difference in recording environments and the influence of environmental sounds may have impacted the results."
- Suggestion: Provide specific examples of how environmental sounds could affect the speech analysis and discuss any measures taken to mitigate this.
-
Line 250-251: "Given that these factors may have affected the results, we would like to increase the data and examine the results after adjusting for the relevant factors in the future."
- Suggestion: Specify which factors (age, gender, severity of AD) will be adjusted for and how this adjustment will be made in future studies.
Author Response

(The authors gave the same response as above.)

Reviewer 3 Report
Comments and Suggestions for AuthorsТhe authors can think of another way to share all the tables, for example, by uploading the results online and including a link in the study. Thus, they will have a volume to describe in detail the differences between the indicators for the three classification models.

Author Response

(The authors gave the same response as above.)

Round 2
Reviewer 1 Report
Comments and Suggestions for Authors
After reviewing the authors' responses to my previous comments on the manuscript submitted to Healthcare Journal, I find their revisions and explanations satisfactory and well-addressed in the following areas:
- Introduction: The authors have incorporated relevant studies on AI-based diagnostic tools for Alzheimer’s disease (AD) and clarified the recent developments in the field. Their addition of references on AD detection through voice analysis strengthens the introduction's foundation and provides necessary context for readers.
- Methods: The revised methodology is more detailed, especially regarding the recording environment, noise removal procedures, and feature selection for speech analysis. This improves clarity and reproducibility. Their addition of specific noise sources, software used for feature extraction, and further details on classifiers (LR, SVM, RF) enhances the methodological rigor and allows readers to understand the approach better.
- Results: The authors have improved figure clarity by explaining the X- and Y-axes in footnotes, making the data representation more accessible. This enhancement supports a better understanding of classification outcomes and error rates in voice-based AD detection.
- Discussion: The authors addressed my concerns about noise interference and sample size limitations. By acknowledging these limitations, the discussion now provides a more balanced view of the study’s findings and potential biases, increasing its transparency and scientific rigor.
- English Language: The authors have rechecked the manuscript with a native English speaker, and their response reflects efforts to improve readability and fluency.
Based on these thorough revisions and explanations, I recommend accepting the manuscript as the authors have successfully addressed all critical feedback, making the study clearer and more robust.
Author Response
We would like to thank you for reviewing our work and for generously evaluating our revisions.
Based on reviewer 2's comments, we have made the necessary additions and revisions. We have also asked a specialist English editing company to review the manuscript again, as the reviewer pointed out that the English was not sufficiently clear. Please see the attached certificate.

Reviewer 2 Report
Comments and Suggestions for Authors
Methodology Details: While the methodology is generally well-described, it would be beneficial to include more details on the selection criteria for participants. For example, they could specify the inclusion and exclusion criteria for both the Alzheimer’s disease (AD) patients and the healthy elderly participants. This might include age ranges, cognitive assessment scores, and any medical history that was considered.
Statistical Analysis: In the Methods section, when discussing the cognitive function assessments, the authors mention the Mini-Mental State Examination (MMSE). It would be beneficial to provide a rationale for choosing this specific test over others. Additionally, if other assessments were used, such as specific speech analysis tools or machine learning algorithms, these should be clearly identified and justified in terms of their relevance to the study objectives.
Language and Style: Review the manuscript for any areas where the language could be simplified or clarified. For example, phrases like "the spectra showed significant differences between the two groups for all components" could be rephrased to "we found significant differences in the speech spectra between the two groups." Simplifying technical language will make the paper more accessible to a wider audience, including those who may not have a strong background in the field.
Comments on the Quality of English LanguageClarity and Conciseness: Overall, the manuscript presents complex ideas, but some sentences could be more concise. For example, long sentences with multiple clauses can be broken down into shorter, clearer statements. This will enhance readability and ensure that key points are easily understood. An example might be a sentence that reads: "The spectra showed significant differences between the two groups for all components, including the mean and standard deviation of the center of gravity, skewness, and kurtosis, and the standard deviation of the spectra."
Grammar and Syntax: There are a few instances of awkward phrasing or grammatical errors that could be improved. A thorough proofreading for grammatical accuracy, including subject-verb agreement and proper tense usage, is recommended. For example, ensure that all verbs are in the correct tense and that plural nouns are matched with plural verbs. A sentence like: "However, the mechanism of that change remains unclear and requires further investigation." and "The recordings of the individuals with AD took place in a hospital environment, whereas the healthy elderly participants were recorded in another setting."
Consistency in Terminology: Ensure that terms are used consistently throughout the manuscript. For instance, if "Alzheimer's disease" is abbreviated as "AD" in one section, it should be consistently referred to as "AD" thereafter, after the first mention. This helps avoid confusion.
Use of Passive Voice: While passive voice is common in scientific writing, it can sometimes make sentences less direct. Where appropriate, consider using active voice to make statements more engaging and clear. For example, instead of saying "The data were analyzed," you could say "We analyzed the data."
Author Response
Thank you for reviewing this manuscript and allowing us to make further revisions.
Based on your comments, we have made the necessary additions and revisions. We have also asked a specialist English editing company to review the manuscript again, as the reviewer pointed out that the English was not sufficiently clear. Please see the attached certificate. The corrections made in response to the comments we received are as follows.
Methodology Details:
Thank you very much for your kind reminder. We have added the following exclusion criteria for the target population.
The exclusion criteria for subjects were the presence of aphasia and difficulty obtaining consent from the patient. We did not apply selection criteria regarding age or gender.
Statistical Analysis:
Thank you for your useful comments on the methods. We have added an explanation of why we used the MMSE and HDS-R. Please note that we did not use any specific speech analysis tool in this study.
We used MMSE and HDS-R for speech recording evaluation because we sought to prioritize simplicity and avoid subject fatigue over a long period of time, rather than for the purpose of diagnosing AD.
Language and Style:
Thank you very much for your very detailed advice on how to express the sentences. We have made the following revisions based on your comments.
We found significant differences in almost all elements of the speech spectrum between the two groups. Regarding intensity, we found significant differences in all factors except standard deviation between the two groups.
Clarity and Conciseness:
Thank you very much for your kind comments on clarity and conciseness, as well as grammar. We have revised the article again, this time requesting a review by a company specializing in English editing, as we believe that this is also a limitation of our English ability. Please check the attached certificate.
Grammar and Syntax:
Thank you very much for your help with syntax. We have made the following corrections.
However, the mechanism of this change was unclear, and further investigation is needed.
The recordings of AD patients were obtained in a hospital environment, whereas the recordings of healthy elderly people were obtained in a different environment.
Consistency in Terminology:
Thank you very much. We have also checked the consistent use of terms again and made corrections.
Use of Passive Voice:
Thank you very much for your carefulness. We have also had the use of the passive voice checked by a Native American and made corrections wherever possible.
